# Self-reported impact of the COVID-19 pandemic, affective responding, and subjective well-being: A Swedish survey

**Maria Gröndal**[ID]*, **Karl Ask**[ID], **Timothy J. Luke**[ID], **Stefan Winblad**

Department of Psychology, University of Gothenburg, Gothenburg, Sweden

* maria.grondal@psy.gu.se

**Data Availability Statement:** All relevant data files are available from the OSF database at the following URL: https://osf.io/f8y3n/?view_only=18297f197c604f94aaf6748328d3cbfd.

## Abstract

A rapid stream of research confirms that the COVID-19 pandemic is a global threat to mental health and psychological well-being. It is therefore important to identify both hazardous and protective individual factors during the pandemic. The current research explored the relationships between self-reported affective responding, perceived personal consequences of the COVID-19 pandemic, and subjective well-being. An online survey ($N$ = 471) conducted in Sweden between June and September, 2020, showed that higher levels of irritability, impulsivity, and the tendency to experience and express anger were generally associated with more severe personal consequences of the pandemic, particularly in areas related to family life, work/study, and finances. While more severe impacts of the pandemic in these areas of life were directly associated with lower subjective well-being, emotion regulation through cognitive reappraisal appeared to moderate the extent to which consequences of the pandemic in other areas of life (i.e., social, free-time and physical activities) translated into decreased well-being. This suggests that cognitive reappraisal may serve to protect against some of the debilitating effects of the COVID-19 pandemic on mental health. Overall, the results indicate that the perceived consequences of the pandemic are multifaceted and that future research should examine these consequences using a multidimensional approach.

## Introduction

The new coronavirus (SARS-CoV-2) and its associated disease (COVID-19) are having a massive impact on public health and have, since their emergence in 2019, changed the daily life of people worldwide [1]. Countries have implemented measures to prevent the virus from spreading. The measures generally include restrictions and recommendations to minimize exposure to the virus by making people maintain a safe physical distance to others, wear face masks, and maintain good personal hygiene [2]. Virus prevention restrictions, especially those relating to social distancing, affect various life domains such as family and social life, work and studies, free time, and physical activities, which may generate both short- and long-term negative consequences for public mental health [3–5].

**Funding:** The authors received no specific funding for this work.

**Competing interests:** The authors have declared that no competing interests exist.

The research on COVID-19 and mental health is rapidly evolving. Scientists have identified numerous ways in which the social and behavioral sciences can support humanity's response to the pandemic, including reducing its negative emotional impact and facilitating adaptive stress management and coping strategies [6]. Moreover, because of its extraordinary global impact, rapid attempts have been made to develop instruments specifically tailored to capturing the unique psychological consequences and perceptions of COVID-19 [e.g., 7, 8]. Although the field is currently dominated by cross-sectional studies, with limited evidence on long-term developments, overall patterns have become apparent in the literature. Specifically, a review published in May 2020 indicated that symptoms of depression and anxiety had increased, and that well-being had decreased, among the general population since the outbreak of the pandemic (the World Health Organization declared COVID-19 a pandemic in March 2020) [9]. A majority of the studies included in the review had been conducted in China, but a similar development has been found also in Western countries [10–12]. A high subjective well-being has also been found to have a protective role against negative reactions (e.g., in terms of distress and depression) to the pandemic [5] and recent findings suggest that higher levels of perceived meaning in life and life satisfaction correlate with lower anxiety and stress related to COVID-19 [13].

Well-being is generally defined as a multi-faceted construct, including both affective and cognitive aspects [14] and is influenced by a person's psychological, social and physical resources as well as by the life challenges the person faces [15]. The COVID-19 pandemic has forced many people into new challenges and uncertainty about the future, which likely impact their well-being. Nevertheless, it would be too wide of a generalization to assume that everyone is negatively mentally affected by the pandemic [16]. Most likely, there is substantial variance in adjustment, where some groups are more vulnerable than others. Therefore, it is of high interest to identify psychological factors that can have a crucial role for the mental outcome for a person during the pandemic, including both protective and impairing aspects.

The ability and flexibility to effectively regulate emotional responses have in recent decades been shown to be related to well-being [17–19]. However, this has mostly been studied in a stable setting with a long-term perspective and not during extreme circumstances or rapidly changing contexts like pandemics [20]. People use emotions in their daily life to interpret and handle situations and it has been suggested that the individual's evaluation of the situation is more important for the emotional reaction than the event itself [21, 22]. Overall, people are likely to feel stronger and more negative emotions during a crisis like the COVID-19 pandemic, because it puts people into new challenging and stressful situations such as managing new family routines or abruptly adjusting to work or study from home [23]. Beyond their consequences for individuals' well-being, emotional responses to the COVID-19 pandemic also seem to play a role in individuals' willingness to take protective action [e.g., 24, 25].

Two commonly used emotion regulation strategies are the antecedent-focused strategy *cognitive reappraisal* and the response-focused strategy *expressive suppression* [19]. The cognitive reappraisal strategy is generally considered adaptive and refers to an early-stage evaluation and reframing of a situation in order to downregulate negative emotions. This strategy is associated with higher well-being, lower negative affect [26] and better adjustment to stress [21, 27, 28]. Current research also indicates that cognitive reappraisal seems to have an adaptive role in reducing stress and anxiety during COVID-19 isolation [29]. Expressive suppression, on the other hand, refers to the inhibition of inner emotions from being expressed outwardly and is categorized as a maladaptive avoidant strategy associated with prolonged negative emotions and, in a long-term perspective, an increase of anxiety and depression [19]. Recent findings indicate that extensive use of expressive suppression and other avoidant coping strategies could be a risk factor for perceived negative mental health [12, 30] and increased loneliness

[31] during the COVID-19 pandemic. In the current study, we explore whether the two emotion regulation strategies (cognitive reappraisal and expressive suppression) moderate the relationship between experienced consequences of COVID-19 in different life domains and subjective well-being.

Data for the present study were collected at an early stage of the COVID-19 pandemic (between June 19 and September 1, 2020) in Sweden. Compared with other countries affected by COVID-19, Sweden has remained relatively open throughout the pandemic, without any nationally enforced lockdowns or requirements of mandatory wearing of face masks [32]. Instead, the Swedish authorities advise people to practice social distancing on a voluntary basis, for instance by working from home when possible, studying remotely, reducing unnecessary travelling, and avoiding social gatherings. The comparatively liberal governmental response in Sweden makes the current sample an important complement to previous studies conducted in areas with more restrictive measures in place.

The data were collected as part of a larger study on the measurement of individual differences in irritability. Thus, in addition to exploring the role of emotion regulation strategies, we had the opportunity to examine the role of three additional affective dispositional variables: trait anger [33], impulsivity [34], and irritability [35]. The potential relevance of these traits stems from the fact that they reflect individual differences in reactions to adversity. In fact, an increased risk of dysfunctional responding to negative outcomes has been observed for individuals with higher levels of trait anger [36], impulsivity [37] and irritability [38]. More recently, higher levels of related affective styles (i.e., cyclothymic, depressive, irritable, and anxious) have been linked to a higher risk of psychological distress as a consequence of the COVID-19 pandemic [39].

Using an explorative approach, the aim of the current study was to examine the relationships between, on one hand, perceived consequences of the pandemic across different life domains and, on the other hand, well-being, emotion regulation strategies, emotional disposition variables (irritability, anger, and impulsivity), and demographics. Although we had not made specific predictions prior to the collection of the data, we were particularly interested in exploring whether emotion regulation strategies play a protective or aggravating role regarding the relationship between self-reported impacts of the COVID-19 pandemic and subjective well-being.

## Method

### Participants and statistical power

Participants were recruited from the research participant pool at the Department of Psychology, University of Gothenburg, Sweden, from a national online platform for research participant recruitment (https://www.studentkaninen.se/), and through announcements at a public library. A total of 576 participants completed the questionnaire, 105 (18.2%) of whom were excluded according to the exclusion criteria (see Procedure and Materials). The final sample thus consisted of 471 participants, 355 (75.4%) of which were women, 112 (23.8%) men, and 4 (0.8%) identifying with other genders. Participants' average age was 33.16 years ($SD = 11.27$, $Mdn = 31$, min = 18, max = 75). In terms of occupational status, 372 participants (79.0%) were employed or studying, 52 (11.0%) unemployed, 25 (5.3%) on parental or medical leave, 10 (2.1%) retired, and 12 (2.5%) "other". Participants had an average of $M = 15.23$ years of education ($SD = 3.29$, $Mdn = 15$).

A sensitivity analysis conducted using G*Power 3.1 [40] indicated that our final sample size ($N = 471$) offered 80% power to detect a bivariate correlation of $r = .128$, and 99% power to detect $r = .194$, at the .05 significance level.

## Procedure

Data for this study were collected between June 19 and September 1, 2020, as part of a larger study on the measurement of irritability. That study was approved by Jesper Lundgren, Head of the Department of Psychology, University of Gothenburg and the preregistration is available at [https://osf.io/f9hg2/]. Participants received an email invitation containing brief information about the study and a link to the survey hosted on the Qualtrics online survey platform (https://www.qualtrics.com). On the first page of the survey, participants were informed that their participation was voluntary, that they had the right to withdraw from the study without any consequences, and that analyses of their responses would not be presented at the individual level. Participants received 40 SEK ($\approx$ 4 EUR) for their participation, which lasted on average 24 minutes ($SD$ = 19 min). That study was approved by Jesper Lundgren, Head of the Department of Psychology, University of Gothenburg and the preregistration is available at.

## Measures

In this section we report only the measures that were analyzed for the current study. The full questionnaire can be found at [https://osf.io/f9hg2/]. The first five instruments below were presented to participants in randomized order, whereas the ratings of COVID-19 impact and the demographic variables were presented at the end of the questionnaire for all participants, in that order [41].

**Brief Irritability Test.** The Brief Irritability Test (BITe) is a five-item scale measuring irritability—"a proneness to experience negative affective states, such as anger, annoyance, and frustration upon little provocation" [35]. Participants rated the frequency with which they had experienced each of five expressions of irritability (e.g., "I have been grumpy") in the last two weeks, using a scale from 1 (*never*) to 6 (*always*). The ratings were summed across items to form a total BITe score between 5 and 30, with higher scores indicating higher irritability. The scale was translated to Swedish by the authors in collaboration with two bilingual, native English speakers. The BITe scale displayed strong internal reliability in the current sample (Cronbach's $\alpha$ = .91, $\omega_{total}$ = .91).

**Emotion Regulation Questionnaire.** The Emotion Regulation Questionnaire [ERQ; 19] measures individual differences in two prominent strategies for the regulation of affective experiences: reappraisal and suppression. In the current study, the Swedish translation of the ERQ developed by Enebrink et al. [42] was used. Participants rated a total of 10 statements on a seven-point scale (1 = *strongly disagree*, 4 = *neutral*, 7 = *strongly agree*). The *reappraisal* subscale consists of six items (e.g., "I control my emotions by changing the way I think about the situation I'm in") and measures the tendency to construe situations in ways that change its emotional impact ($\alpha$ = .88, $\omega_{total}$ = .88). The *suppression* subscale consists of four items (e.g., "I keep my emotions to myself.") and measures the tendency to inhibit ongoing emotion-expressive behavior ($\alpha$ = .76, $\omega_{total}$ = .77). Composite scores for the subscales were computed by summing the respective items so that higher scores indicated a higher degree of reappraisal and suppression, respectively.

**State-Trait Anger Expression Inventory-2.** The State-Trait Anger Expression Inventory-2 [STAXI-2; 43] is a revised and extended version of the original STAXI scale [33]. It measures three aspects of anger: experience, expression, and control. For the current study, the Swedish translation developed by Lindqvist et al. [44] was used. Participants rated a total of 57 items using 4-point rating scales (1 = *not at all/almost never*, 4 = *very much/almost always*). The *state anger* subscale consists of 15 items (e.g., "I feel angry") measuring current experience of anger (ordinal $\alpha$ = .97, ordinal $\omega_{total}$ = .97). The *trait anger* subscale consists of 10 items (e.g., "I am quick tempered") measuring the general disposition to experience and express anger (ordinal

α = .90, ordinal $\omega_{total}$ = .89). The remaining subscales are made up of eight items each: *Anger expression-out* (AX-O; e.g., "I express my anger") measures the tendency to express anger toward other persons or objects (ordinal α = .80, ordinal $\omega_{total}$ = .81); *anger expression-in* (AX-I; e.g., "I keep things in") measures the tendency to hold in or suppress angry feelings (ordinal α = .83, ordinal $\omega_{total}$ = .82); *anger control-out* (AC-O; e.g., "I control my temper") measures the tendency to inhibit the expression of angry feelings toward others or objects (ordinal α = .75, ordinal $\omega_{total}$ = .76); and *anger control-in* (AC-I; e.g., "I take a deep breath and relax") measures the tendency to control angry feelings by calming down (ordinal α = .88, ordinal $\omega_{total}$ = .88). For each subscale, the respective items were summed to form a composite score.

**Satisfaction With Life Scale.** The Satisfaction With Life Scale [SWLS; 45] is a brief measure of subjective well-being. A Swedish translation of the SWLS [46, 47] was used in the current study. Participants rated their agreement with five statements (e.g., "In most ways my life is close to my ideal") using a seven-point scale (1 = *strongly disagree*, 7 = *strongly agree*). The individual ratings were summed to form a total SWLS score between 5 and 35, with higher scores reflecting higher satisfaction with life. In the current sample, the internal consistency of the items was high (α = .91, $\omega_{total}$ = .91).

**UPPS Impulsive Behavior Scale.** The UPPS Impulsive Behavior Scale [34] consists of four subscales measuring different facets of trait impulsivity: urgency, (lack of) premeditation, (lack of) perseverance, and sensation seeking. A Swedish translation of the UPPS scale developed by Marklund [48] was used in the current study. Participants rated their agreement with a total of 45 statements using a four-point scale (1 = *agree strongly*, 4 = *disagree strongly*). The *urgency* subscale consists of 12 items (e.g., "I have trouble controlling my impulses") and measures the tendency to engage in impulsive behaviors in order to alleviate negative affect (ordinal α = .92, ordinal $\omega_{total}$ = .92). The *(lack of) premeditation* subscale consists of 11 items (e.g., "I am a cautious person") and measures the tendency not to reflect or deliberate on the consequences of behaviors before engaging in them (ordinal α = .88, ordinal $\omega_{total}$ = .88). The *(lack of) perseverance* subscale consists of 10 items (e.g., "I finish what I start") and measures the inability to remain focused on a difficult or boring task and to resist distractions (ordinal α = .82, ordinal $\omega_{total}$ = .85). Finally, the *sensation seeking* subscale consists of 12 items (e.g., "I'll try anything once") and measures an individual's openness to trying risky, exciting activities and tendency to enjoy such activities (ordinal α = .89, ordinal $\omega_{total}$ = .89). For each subscale, a total score was calculated by summing the individual item ratings (after reverse scoring where appropriate), so that higher scores meant higher impulsivity.

**COVID-19 impact.** Six items were created for the present study to measure the perceived impact of the COVID-19 pandemic on different domains of participants' lives. Participants were asked to rate how and to what extent the quality of their family life, work/studies, finances, social life, free time activities, and physical activity had changed because of the pandemic, using a scale from 1 (*has become much worse*), via 4 (*has not been affected*), to 7 (*has become much better*).

**Demographic variables.** At the very end of the questionnaire, participants were asked to report their year of birth, their gender (*female*, *male*, or *other*), their current occupation (*employed/student*, *unemployed*, *medical/parental leave*, *retired*, or *other*), and total number of years of education.

**Attention checks.** Three attention checks were included in the questionnaire to enable the exclusion of participants who failed to pay attention to the question content. On the page directly following the BITe, the UPPS, and the STAXI-2, respectively, participants were presented with five options and asked to select the one option that best described what the questions on the previous pages were about. As an example, the response options presented

following the BITe were "irritability" (correct answer), "religious belief", "cognitive function-ing", "exercise habits", and "sadness".

## Exclusion criteria

Participants were excluded from all data analyses (a) if they had completed the survey in less than 5 minutes ($n = 0$) and/or (b) if they failed to correctly answer any of the attention checks ($n = 87$). These criteria were part of the preregistration of the original study [https://osf.io/f9hg2/]. Moreover, participants were excluded if they had participated more than once ($n = 16$) and if they were under 18 years of age ($n = 2$). The combined criteria led to the exclusion of 105 participants.

# Results

## How do self-reported impacts of COVID-19 relate to demographics?

Generally, there were no significant correlations between demographic variables and the self-reported impacts of COVID-19. One exception concerned participants' occupation. To examine the relationship between occupation and each of the six self-reported COVID-19 impacts, we fit a series of regression models in which occupation (dummy coded with employed/study-ing as reference group) predicted the pandemic impact items. These models indicated that, compared to those who were employed or studying, people who were unemployed reported being significantly more negatively impacted by COVID-19 in their work and studies (work-ing/studying $M = 3.32$, $SD = 1.60$ vs. unemployed $M = 2.11$, $SD = 1.34$), $b = -1.20$, 95% [-1.65, -0.75], $t(466) = 5.24$, $p < .001$, and in their finances (working/studying $M = 3.76$, $SD = 1.30$ vs. unemployed $M = 2.40$, $SD = 1.47$), $b = -1.35$, 95% [-1.73, -0.98], $t(466) = 7.13$, $p < .001$. For more complete information, including all nonsignificant results, see the online supplemental materials [https://osf.io/f8y3n/].

## How do self-reported impacts of COVID-19 relate to individual differences?

We examined the correlations between the six COVID-19 impact items and the individual dif-ference measures (see Table 1). Several notable patterns emerged. Scores on the BITe were sig-nificantly negatively correlated with several self-reported impacts of the pandemic, such that people who reported being more irritable also reported more negative impacts on family life, work and study, and finances. The UPPS sensation-seeking subscale was significantly posi-tively correlated with pandemic impacts on social life and free time activities, such that people scoring higher on sensation seeking reported less negative impacts in those domains. People who scored higher on the UPPS (lack of) perseverance subscale tended to report significantly more negative impacts on work and study and on finances. Those scoring higher on the UPPS urgency subscale reported significantly more negative impacts on family life and finances.

Furthermore, individual differences in the expression and control of anger were associated with the reported impact of COVID-19. Specifically, people who scored higher on the anger control-out subscale of the STAXI (i.e., those more likely to inhibit outward expressions of anger) tended to report significantly less negative impacts on family life. Conversely, those who scored higher on the anger expression-out subscale (i.e., those more likely to express their anger toward others) tended to report more negative impacts on family life. People who scored higher on the anger expression-in subscale of the STAXI (i.e., those more likely to suppress or hold in angry feelings) tended to report significantly more negative impacts on work/study and on finances.

**Table 1. Correlations between individual difference measures and self-reported COVID-19 impacts.**

| | COVID-19 impact | | | | | |
| --- | --- | --- | --- | --- | --- | --- |
| | **Family life** | **Work and study** | **Finances** | **Social life** | **Free time activities** | **Physical activity** |
| BITe | **-.124 [-.212, -.034]** | **-.100 [-.189, -.010]** | **-.147 [-.235, -.058]** | .020 [-.070, .111] | -.005 [-.095, .085] | -.074 [-.163, .017] |
| ERQ R | .016 [-.075, .106] | .052 [-.039, .141] | .048 [-.043, .138] | -.035 [-.125, .056] | -.016 [-.106, .074] | .053 [-.037, .143] |
| ERQ S | .009 [-.082, .099] | -.019 [-.109, .072] | -.001 [-.092, .089] | -.033 [-.123, .058] | -.036 [-.126, .054] | -.011 [-.101, .079] |
| STAXI state | -.077 [-.166, .013] | -.027 [-.117, .064] | **-.121 [-.209, -.031]** | .062 [-.029, .151] | .022 [-.069, .112] | -.016 [-.106, .075] |
| STAXI trait | **-.141 [-.229, -.052]** | -.068 [-.158, .022] | -.085 [-.174, .005] | .000 [-.091, .090] | .034 [-.056, .124] | -.042 [-.132, .048] |
| STAXI AC-I | .021 [-.069, .111] | -.068 [-.157, .023] | -.063 [-.153, .027] | -.060 [-.150, .030] | -.043 [-.133, .047] | .066 [-.025, .155] |
| STAXI AC-O | **.114 [.024, .202]** | .086 [-.004, .175] | .046 [-.045, .135] | .024 [-.066, .114] | .033 [-.058, .123] | .074 [-.016, .164] |
| STAXI AX-I | -.049 [-.138, .042] | **-.093 [-.182, -.003]** | **-.095 [-.184, -.005]** | -.021 [-.111, .069] | -.002 [-.092, .089] | .013 [-.077, .104] |
| STAXI AX-O | **-.143 [-.231, -.054]** | -.053 [-.142, .038] | -.066 [-.156, .024] | .022 [-.069, .112] | .013 [-.077, .103] | -.030 [-.120, .061] |
| SWLS | **.115 [.025, .203]** | **.185 [.096, .270]** | **.195 [.107, .281]** | .069 [-.022, .158] | .030 [-.061, .120] | .057 [-.034, .146] |
| UPPS PM | .028 [-.062, .118] | .013 [-.077, .103] | .035 [-.055, .125] | .054 [-.036, .144] | .003 [-.088, .093] | .058 [-.033, .148] |
| UPPS PS | -.055 [-.145, .035] | **-.121 [-.209, -.031]** | **-.136 [-.224, -.046]** | -.016 [-.106, .075] | -.090 [-.179, .000] | -.038 [-.128, .053] |
| UPPS S | .084 [-.006, .174] | .045 [-.046, .134] | .040 [-.051, .130] | **.123 [.033, .211]** | **.094 [.004, .183]** | .080 [-.010, .169] |
| UPPS U | **-.100 [-.188, -.009]** | -.032 [-.122, .059] | **-.120 [-.208, -.030]** | .051 [-.039, .141] | -.016 [-.106, .075] | -.044 [-.134, .047] |

Note: Correlation coefficients are displayed with 95% CIs. Significant correlations are displayed in boldface. BITe = Brief Irritability Scale; ERQ R = Emotion Regulation Questionnaire reappraisal subscale; ERQ S = Emotion Regulation Questionnaire suppression subscale; STAXI state = STAXI state subscale; STAXI trait = STAXI trait subscale; STAXI AC-I = STAXI anger control-in subscale; STAXI AC-O = STAXI anger control-out; STAXI AX-I = STAXI anger expression-in; STAXI AX-O = STAXI anger expression-out; SWLS = Satisfaction with Life Scale; UPPS PM = UPPS (lack of) premeditation subscale; UPPS PS = UPPS (lack of) perseverance subscale; UPPS S = UPPS sensation seeking subscale; UPPS U = UPPS urgency subscale.

Finally, and perhaps unsurprisingly, the SWLS was positively correlated with self-reported impacts of the pandemic on family life, work and study, and finances. That is, people who experienced more serious impacts of COVID-19 in these domains tended to be less satisfied with their life.

## How do self-reported impacts of COVID-19 relate to each other?

Descriptive statistics for the six self-report items concerning COVID-19 are displayed in Table 2. The means for all items were below the midpoint of the scale, suggesting that people on average found all measured domains of their lives somewhat negatively impacted by the pandemic. As can be seen in Table 2, the six COVID-19 impact measures were all significantly positively correlated with each other. The strongest correlations were found between domains

**Table 2. Descriptive statistics and correlations between self-reported COVID-19 impacts.**

| | **Family life** | **Work and study** | **Finances** | **Social life** | **Free time activities** | **Physical activity** |
| --- | --- | --- | --- | --- | --- | --- |
| Family life | | .271 [.185, .352] | .241 [.154, .324] | .231 [.144, .315] | .257 [.170, .339] | .197 [.108, .282] |
| Work and study | | | .482 [.410, .549] | .300 [.215, .380] | .260 [.174, .343] | .244 [.157, .327] |
| Finances | | | | .187 [.098, .273] | .202 [.113, .287] | .208 [.120, .293] |
| Social life | | | | | .587 [.525, .644] | .330 [.247, .408] |
| Free time activities | | | | | | .499 [.428, .564] |
| *Mean* | 3.49 | 3.16 | 3.56 | 2.87 | 3.04 | 3.67 |
| *SD* | 1.24 | 1.60 | 1.35 | 1.27 | 1.26 | 1.48 |
| *Median* | 3 | 3 | 4 | 3 | 3 | 4 |

*Note.* Correlation coefficients are displayed with 95% CIs. All items measured on a 1–7 scale (1 = *much worse*, 4 = *not affected*, 7 = *much better*).

that are functionally related, such as work/study–finances, social life–free time activities, and free time activities–physical activity.

For subsequent analyses, we were interested in examining whether the six COVID-19 impact items could be meaningfully grouped into composite variables. To assess the structure of the relationships between the items, we conducted an exploratory factor analysis on the six items, conducted using the *psych* package [49] in R. A parallel analysis suggested that two factors should be retained. Thus, we extracted two factors, which explained 26% and 20% of the variance, respectively, and examined the loadings from the oblimin rotated factor matrix (see Table 3). The two factors were composed as follows: (1) social life, free time activities, and physical activities; and (2) family life, work and study, and finances. It should be noted, however, that family life loaded only modestly on the second factor.

## Do emotion regulation strategies moderate the relationship of COVID-19 impacts and satisfaction with life?

Given that at least some of the COVID-19 impact items positively correlated with SWLS scores, we were interested in examining the possibility that people's emotional regulation strategies (as assessed by the ERQ's suppression and reappraisal subscales) moderate this relationship. We calculated sum scores for Factor 1 (family, work and study, and finances) and Factor 2 (social life, free time activities, physical activity) of the self-reported COVID-19 impacts, ERQ reappraisal subscale, ERQ suppression subscale, and SWLS.

In a first regression analysis, we assessed whether COVID-19 impact Factor 1, ERQ reappraisal subscale, ERQ suppression subscale, and the ERQ subscales' interactions with Factor 1 predicted SWLS scores. We found that the addition of both interactions terms in a second step of the analysis did not significantly improve model fit over a model without interactions, $F(2, 465) = 2.87$, $p = .058$, but adding only the interaction term between the reappraisal scale and Factor 1 did significantly improve model fit, $F(2, 465) = 5.70$, $p = .017$. In this model (see Table 4, upper panel), reappraisal and suppression significantly predicted SWLS, and the interaction of Factor 1 and reappraisal was significant, such that for those who scored lower on reappraisal, Factor 1 predicted SWLS to a greater extent. A Johnson-Neyman analysis indicated that the coefficient for COVID-19 Factor 1 was significant and positive at ERQ reappraisal values at least 2.91 points below the sample mean (see Fig 1). In other words, negative effects of COVID-19 on social life, free time activities, and physical activity predicted decreased satisfaction with life for those scoring below average on the use of reappraisal strategies. The *p*-value for this interaction was relatively high (.017), so it should be interpreted with caution. As a robustness check, we tested an analogous structural equation model using latent variables for each of the scales. In this model, the interaction between Factor 1 and the ERQ reappraisal subscale was also significant, $p = .026$. For further details on this analysis, see https://osf.io/f8y3n/.

**Table 3. Factor loadings of self-reported COVID-19 impacts.**

|  | Factor 1 | Factor 2 |
|---|---|---|
| Family life | 0.17 | **0.31** |
| Work and study | 0.02 | **0.74** |
| Finances | -0.03 | **0.65** |
| Social life | **0.56** | 0.14 |
| Free time activities | **0.97** | -0.04 |
| Physical activity | **0.46** | 0.15 |

*Note*. Loadings from oblimin rotation. Loadings greater than 0.30 are displayed in boldface.

**Table 4. Results of regression models predicting satisfaction with life from self-reported COVID-19 impact and emotion regulation strategies.**

| Predictor | Unstandardized estimate (b) [95% CI] | p-value |
|---|---|---|
| Factor 1 | | |
| COVID-19 impact Factor 1 | 0.117 [-0.069, 0.303] | .213 |
| ERQ R | 0.314 [0.230, 0.398] | < .001 |
| ERQ S | -0.321 [-0.444, -0.198] | < .001 |
| COVID-19 impact Factor 1 × ERQ R | -0.026 [-0.047, -0.005] | .017 |
| Factor 2 | | |
| COVID-19 impact Factor 2 | 0.474 [0.280, 0.668] | < .001 |
| ERQ R | 0.289 [0.207, 0.372] | < .001 |
| ERQ S | -0.330 [-0.451, -0.208] | < .001 |

*Note.* ERQ R = Emotion Regulation Questionnaire reappraisal subscale; ERQ S = Emotion Regulation Questionnaire suppression subscale.

In a second regression analysis, we found that COVID-19 impact Factor 2 and the ERQ subscales significantly predicted SWLS, but adding the subscales' interaction terms with Factor 2 in a second step did not significantly improve the fit of the model, $F (2, 465) = 0.18$, $p = .83$. The results of the retained model, without the interaction terms, are displayed in Table 4 (lower panel).

## Discussion

Recent research seems to converge on the fact that the COVID-19 pandemic has confronted people worldwide with new challenges in daily life, leading to negative consequences for public mental health. Thus, it is of high importance to examine and identify both hazardous and protective factors for individuals during the pandemic. The current study examined (a) the extent to which individual differences in affective responding are associated with reported consequences of the COVID-19 pandemic in different life domains, and (b) the extent to which emotion regulation strategies—cognitive reappraisal and expressive suppression—moderate the relationship between reported consequences of COVID-19 and subjective well-being.

The results showed few relationships between demographic variables and reported consequences of COVID-19, the only exception being that unemployed respondents reported having suffered more severe consequences for their work/study life and finances. This is perhaps not a surprising result since the labor market during the pandemic has been strained. Many people have lost their jobs as a consequence of the pandemic, and people who were already unemployed when the pandemic started are having a harder time to get an employment. The negative consequences of increased unemployment caused by the COVID-19 pandemic may become a serious public health problem, which in the long term can lead to decreased quality of life [50].

Overall, weak correlations were found between reported consequences of COVID-19 and affective trait and state measures. Given the statistical precision in our estimates, we can somewhat confidently rule out very substantial relationships between these variables. Some exceptional notable patterns emerged, however. First, the affective trait measures were related primarily to reported consequences of the COVID-19 pandemic on their family life, their work or study life, and their finances. Specifically, respondents with a higher tendency to experience and express anger, and those with a higher level of impulsivity, were more likely to report a negative impact on their family life, their work or study life, and their finances. Of

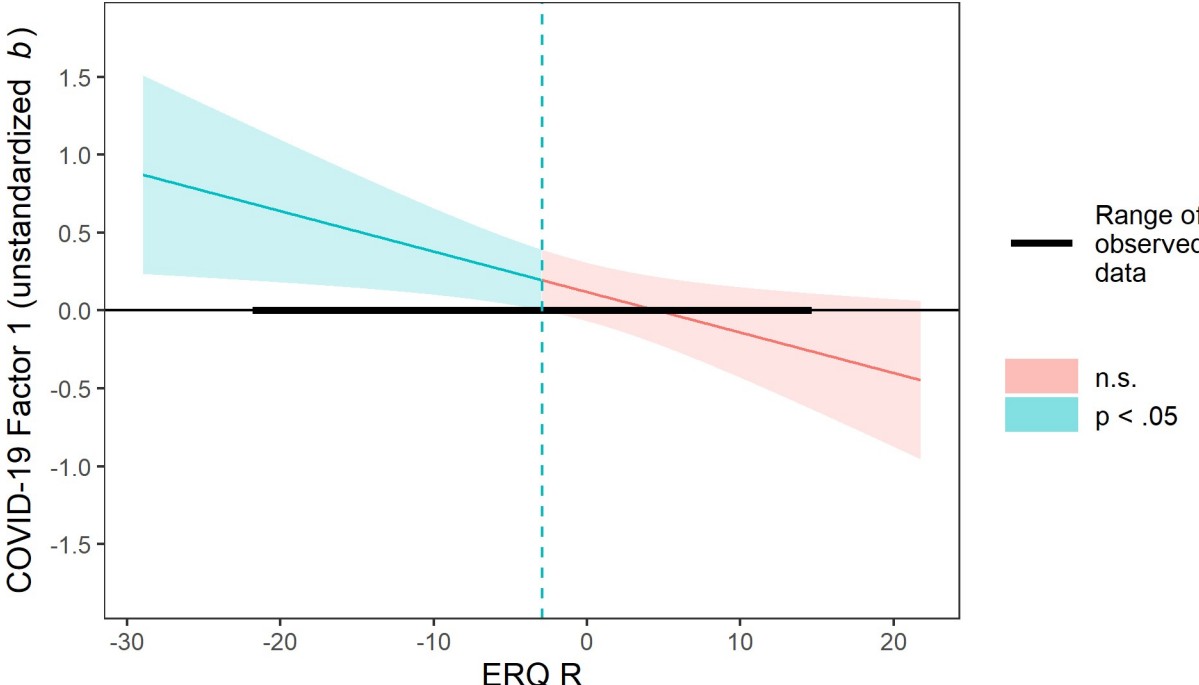

**Fig 1. Conditional effect plot for the interactive effect of COVID-19 impact Factor 1 (social life, free time activities, physical activity) and reappraisal on reported satisfaction with life.** *Note.* COVID-19 impact Factor 1 and ERQ reappraisal scores have been mean centered on this plot and in the corresponding regression model.

particular note, respondents who experience and outwardly express anger (or fail to control such expressions) reported more severe consequences of the pandemic for their family life. These results are consistent with the rising concerns about increasing domestic violence as a direct result of social isolation [51] and previous findings of a relationship between anger and relational problems during the COVID-19 pandemic [52].

The second exception to the overall pattern in our result was that the reported impact of the COVID-19 pandemic on social life and free time activities were positively correlated with sensation seeking, which indicates that high sensation seekers are less negatively affected in those domains. Speculative explanations of this result are that sensation seekers to a higher extent engage in activities that are less affected by restrictions due to COVID-19 (e.g., outdoor activities) and/or that they may be more likely to disregard recommendations for restrictions because of a lower sensitivity to risk [34]. Another possible explanation, supported by previous research, is that higher sensation seeking (in contrast to the other impulsivity subscales used in this study) has been associated with higher psychological well-being [53] and psychological resilience with an increased ability to manage adversity [54]. In other words, it could be that sensation seekers to a higher extent find ways to cope with the new situation caused by the pandemic and manage to make something positive out of it (e.g., by trying new "pandemic-friendly" activities or by finding innovative ways to keep the social life intact though the restrictions). The fact that different patterns were found across the four subscales of the impulsivity scale (UPPS-P) supports the notion that impulsivity is best conceptualized as a multidimensional construct [55].

The current results showed that higher levels of irritability were associated with more severe impacts of the COVID-19 pandemic on family life, work/study, and finances. Given the correlational nature of our data, it is not possible to discern whether the observed relationships

reflect a causal sequence in either direction. First, it may be that individuals with a disposition to experience less irritability are better equipped to deal with the challenges posed by the pandemic (e.g., because they possess more functional coping strategies). Second, it may be that those who suffer more severe consequences of the pandemic experience more irritability as direct result of those consequences. The BITe instrument measures irritability in the past two weeks, so there is some ambiguity as to whether the observed scores mainly reflect a stable disposition or temporary fluctuations in irritability. A third possibility is that irritability (as well as anger and impulsivity, see above) reflects a more general vulnerability factor [e.g., neuroticism; see 31] which is more directly related to the individual's ability to cope with the pandemic. Regardless of which interpretation is more valid, however, one can expect an overall increase in irritability in the general population as a result of the pandemic, since irritability can be elevated by stress, goal disruption (e.g., frustration) or chronic environmental stressors [56, 57].

The reported impact of the pandemic on family life, work/study, and finances was directly associated with subjective well-being, such that respondents who reported more severe consequences of the COVID-19 pandemic in these areas were less satisfied with their life. On the other hand, the relationship between the impact on social life, free time activities, physical activity and subjective well-being appeared to be moderated by respondents' disposition for cognitive reappraisal. For respondents who engage in relatively little cognitive reappraisal, more negative consequences of the COVID-19 pandemic in these areas were associated with lower satisfaction with life. In contrast, for relatively high reappraisers, the impact of COVID-19 on these domains of life was not significantly related to general life satisfaction. This tentative finding suggests that emotion regulation through cognitive reappraisal may protect against the debilitating effects of the COVID-19 pandemic in some areas of life. This potentially protective role of cognitive reappraisal is consistent with previous research demonstrating that this strategy is effective in downregulating negative emotions during stressful events [21, 27, 58]. However, a recent study examining the role of emotion regulation strategies and psychological and physical health during COVID-19 found no relation between cognitive reappraisal and psychological health [30]. Low et al. argued that these results could be because of the pandemic's unpredictable and uncontrollable nature which complicates the possibility to reframe a situation in less negative terms.

Given the exploratory nature of our analyses, combined with the fact that the observed interaction effect barely fell below the .05 significance threshold, we urge for caution when interpreting the potential moderating effect of cognitive reappraisal. However, we believe it is a finding worth pursuing in future confirmatory research. A potential underlying mechanism is the ease with which negative consequences in different domains of life can be reappraised in non-threatening terms. For instance, it may be difficult to reconstrue the loss of a job or the fact of not having enough money for rent in anything but negative terms, because they constitute objective indicators of loss. In contrast, social and leisure activities largely lack such objective indicators and may thus be more malleable to subjective interpretation. In other words, the attempt to reappraise these latter aspects of life may be a more fruitful endeavor. This finding further supports the argument that no emotion regulation strategy is always robustly adaptive. In some stressful situations, other actions than downregulating emotions are likely to be more effective [e.g., to use problem-focused strategies when one has lost one's job; 58]. Should this finding replicate in future studies, it would have important clinical implications, as it would highlight more and less suitable target areas for cognitive restructuring in therapeutic interventions. Such knowledge is of great value as the accumulated consequences of the COVID-19 pandemic on mental health are likely to unfold in the near future.

## Limitations

A few limitations of the current study should be noted. First, the analyses in this study were exploratory without prior hypotheses, and the results should therefore be interpreted with some caution. Nonetheless, we think that the novel findings discussed above raise possibilities worth addressing in future confirmatory research. Second, the correlational design of this study prevents causal interpretations of the results. Hence, our speculations regarding mechanisms underlying the findings in this study should be considered preliminary and in need of validation using alternative methods (e.g., longitudinal designs). Third, the extent to which the present findings are specific to the geographic (Sweden) and temporal (the summer of 2020) context present during the data collection is unclear. While these idiosyncrasies may prove important for demonstrating cultural and temporal variations in psychological responses to the COVID-19 pandemic, potential contextual differences need to be considered in any attempts to generalize the findings across times and settings.

## Conclusions

Taken together, our findings indicate that the impact of the COVID-19 pandemic on individuals' everyday life is multifaceted and needs to be measured multidimensionally. The findings also suggest that individuals with certain psychological characteristics (i.e., those predisposed to irritability, anger, and impulsivity) may be at slightly higher risk of experiencing negative consequences of the pandemic. In contrast, emotion regulation through cognitive reappraisal may serve as protection against negative consequences of the pandemic, particularly in areas of life related to free-time, physical, and social activities. This finding strongly indicates that it is worthwhile to continue examine the role of emotion regulation strategies in relation to well-being during the COVID-19 pandemic. Understanding the adaptiveness of different emotion regulation strategies may prove crucial to preventing declines in subjective well-being during pandemics and other disruptive global events.

## Acknowledgments

We would like to thank all the respondents who were willing to answer this online survey.

## Author Contributions

**Conceptualization:** Maria Gröndal, Karl Ask, Stefan Winblad.

**Formal analysis:** Timothy J. Luke.

**Investigation:** Maria Gröndal.

**Methodology:** Maria Gröndal, Karl Ask, Stefan Winblad.

**Software:** Maria Gröndal.

**Supervision:** Karl Ask.

**Visualization:** Timothy J. Luke.

**Writing – original draft:** Maria Gröndal, Karl Ask, Timothy J. Luke.

**Writing – review & editing:** Maria Gröndal, Karl Ask, Timothy J. Luke, Stefan Winblad.

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
