## [Decision Letter · Decision Letter 0]

20 Apr 2021

PONE-D-21-05122

Self-reported impact of the COVID-19 pandemic, affective responding, and subjective well-being: A Swedish survey

PLOS ONE

Dear Dr. Gröndal,

Thank you for submitting your manuscript to PLOS ONE. After careful consideration, we feel that it has merit but does not fully meet PLOS ONE’s publication criteria as it currently stands. Therefore, we invite you to submit a revised version of the manuscript that addresses the points raised during the review process.

Please find below the reviewers' comments, as well as those of mine.

We look forward to receiving your revised manuscript.

Kind regards,

Valerio Capraro

Academic Editor

PLOS ONE

Journal Requirements:

2. Peer review at PLOS ONE is not double-blinded (https://journals.plos.org/plosone/s/editorial-and-peer-review-process). For this reason, authors should include in the revised manuscript all the information removed for blind review, including names of universities.

Additional Editor Comments:

I have now collected two reviews from two experts in the field. The reviewers like the paper, but suggest several revisions. Therefore, I would like to invite you to revise your work for Plos One following the reviewers' comments, to which I would like to add two more comments. (i) There has been some research, recently, looking at the effect of emotion on pandemic response (i.e., social distancing, wearing a face mask). Some of these papers are Capraro & Barcelo (2021), Heffner et al. (2020), Pfattheicher et al. (2020). This literature seems related to your paper. (ii) More generally, I think that the "perspective article" on what social and behavioral science can do to support pandemic response, published by Van Bavel et al in Nature Human Behaviour can be a useful general reference.

I am looking forward for the revision.

References

Capraro, V., & Barcelo, H. (2021). Telling people to “rely on their reasoning” increases intentions to wear a face covering to slow down COVID-19 transmission. Applied Cognitive Psychology.

Heffner, J., Vives, M. L., & FeldmanHall, O. (2020). Emotional responses to prosocial messages increase willingness to self-isolate during the COVID-19 pandemic. Personality and Individual Differences, 170, 110420.

Pfattheicher, S., Nockur, L., Böhm, R., Sassenrath, C., & Petersen, M. B. (2020). The emotional path to action: Empathy promotes physical distancing during the COVID-19 pandemic. Psychological Science, 31, 1363-1373.

Van Bavel, J. J., et al. (2020). Using social and behavioural science to support COVID-19 pandemic response. Nature Human Behaviour, 4, 460-471.

Reviewers' comments:

Reviewer's Responses to Questions

**Comments to the Author**

1. Is the manuscript technically sound, and do the data support the conclusions?

Reviewer #1: Yes

Reviewer #2: Partly

2. Has the statistical analysis been performed appropriately and rigorously? 

Reviewer #1: Yes

Reviewer #2: Yes

3. Have the authors made all data underlying the findings in their manuscript fully available?

Reviewer #1: Yes

Reviewer #2: Yes

4. Is the manuscript presented in an intelligible fashion and written in standard English?

Reviewer #1: Yes

Reviewer #2: Yes

5. Review Comments to the Author

Reviewer #1: First of all, congratulate the authors for the manuscript. The subject is highly topical and, despite being an exploratory approach, I consider that this type of study is necessary due to the novelty of the pandemic circumstances of this past year.

I have some suggestions about the manuscript:

Perhaps the title could be revised to better fit the exploratory approach.

In the introduction, the authors could mention other works on the validation of measurement instruments that have recently been developed to evaluate different aspects related to the Covid-19 pendemic. For example, Questionnaire on Perception of Threat from COVID-19.

It is also recommended to check that new references have not been included in the discussion that, previously, were not cited in the introduction section.

Reviewer #2: The manuscript entitled „Self-reported impact of the COVID-19 pandemic, affective responding, and subjective well-being: A Swedish survey” describes a study conducted in Sweden and investigating associations between irritability, impulsive behavior, emotion regulation and self-perceived COVID-19 impact. The study is justified in the light of high need for identification of risk and protective factors against the mental health consequences of the pandemic. Some characteristics of Author’s approach indicate their high methodological standards (e.g. preregistration, sample size calculation, predefines criteria of exclusion, checks of participants’ attention in the on-line form, confidence intervals). However, I have several suggestions which may be used to improve the paper.

#1. The introduction is quite loosely related to the main purpose of the study. Although the general aim of the study – searching for risk factors of worsened adaptation to the pandemic – is justified, the sentences indicating a lack of hypotheses are confusing and methodologically doubtful. If Author(s) did not have any hypotheses, how they determine the independent variables? The design of the study indicates that the Authors were investigating anger, irritability and impulsive behaviors, moderator role of emotion regulation strategies and the relatively new construct: self-reported impact of the COVID-19 pandemic. Thus, in the introduction these variables should be not only included but also put in the context of the psychological responding to the pandemic. Why studying anger and related constructs should be useful in the context o the pandemic? Which is the role of restriction due to the pandemic? How about the particular Swedish context in this matter? In my opinion the introduction should be rewritten in order to include the hypotheses or a model of the study.

#2. I think that the Authors can include an exemplary attention check in order to make clear why relatively large portion of the participants was dropped out from the analysis.

#3. Was the self-reported impact of the COVID-19 hypothesized as a unifactorial variable? At the beginning of the results section the Authors described results separately for each item of the self-reported impact, but later on factor analyses them and conducted analyses on the two factors. In my opinion, the factor analysis should be presented from the very beginning and the rest analyses could be conducted regarding the two factors of the self-reported pandemic impact.

#4. In the discussion try to avoid using technical terms like: factor 1 and factor 2 – please use a labels of these factors.

#5. Since the demographical variables are not the central independent variable sin the study, I think that their role in prediction of the COVID-19 impact should be described more concisely (e.g. without separate sections, but only in descriptive statistics section).

#6. While investigating the moderation, I suggest the Author(s) to focus on the particular method (SEM or regression analysis). The figure attached to the manuscript seems to be distorted (when compared to the figure in the osf project). The power analysis in the manuscript was made for correlations analyses. However, the Author(s) may also included post-hoc power analysis for the interaction detection, e.g. using delta f2 (change of the explained variance between steps in hierarchical regression; please remember to included statistics for regression model when describing the interaction: F, adj. R2, change in R2 due to interactions included in the model, etc.).

6. PLOS authors have the option to publish the peer review history of their article (what does this mean?). If published, this will include your full peer review and any attached files.

Reviewer #1: No

Reviewer #2: No

---

## [Author Response · Author response to Decision Letter 0]

28 Jun 2021

Dear Professor Capraro,

We’d like to thank you for the opportunity to resubmit a revised version of our manuscript. We are grateful for your and the reviewers’ careful reading of our original submission and for the thoughtful and constructive comments. We have done our best to accommodate as many of the suggested revisions as possible, and we feel that the paper has improved significantly as a result. Below, we summarize the revisions that we have made. While most of the suggestions have been worked into the manuscript, on a few occasions we were unable to do so. In those cases, we explain why in our responses below. With these revisions, we hope that our paper now meets the criteria for publication in PLOS ONE.

Sincerely,

Maria Gröndal

Karl Ask

Timothy Luke

Stefan Winblad

Journal requirements

The manuscript and files have been prepared according to the style requirements.

2. Peer review at PLOS ONE is not double-blinded (https://journals.plos.org/plosone/s/editorial-and-peer-review-process). For this reason, authors should include in the revised manuscript all the information removed for blind review, including names of universities.

We now include the name of the university at which participants were recruited. We have also switched out the links to the project pages on OSF, so that these are no longer anonymous.

Editor comments

1. There has been some research, recently, looking at the effect of emotion on pandemic response (i.e., social distancing, wearing a face mask). Some of these papers are Capraro & Barcelo (2021), Heffner et al. (2020), Pfattheicher et al. (2020). This literature seems related to your paper.

We have added a sentence on page 4 on the behavioral consequences of emotional responses to the pandemic: “Beyond their consequences for individuals’ well-being, emotional responses to the COVID-19 pandemic also seem to play a role in individuals’ willingness to take protective action (e.g., Capraro & Barcelo, 2021; Phattheicher et al., 2020).” However, since the focus of our paper is the perceived personal consequences of the pandemic and their relationships with subjective well-being, it was not possible to fit a detailed discussion of such studies in the narrative.

2. More generally, I think that the "perspective article" on what social and behavioral science can do to support pandemic response, published by Van Bavel et al in Nature Human Behaviour can be a useful general reference.

We are thankful for the suggestion and agree that the Van Bavel et al. article provides a suitable reference for the potential contribution from the scientific community. We have added a sentence citing the article in the second paragraph of the Introduction (p. 3): “Scientists have identified numerous ways in which the social and behavioral sciences can support humanity’s response to the pandemic, including reducing its negative emotional impact and facilitating adaptive stress management and coping strategies (Bavel et al., 2020).”

Reviewer #1

1. Perhaps the title could be revised to better fit the exploratory approach.

We have scrutinized the title for elements that may suggest that this is not exploratory research, but have not found any. The title lists the constructs that we have measured (self-reported impact of the pandemic, affective responding, and subjective well-being) and notes that it is a survey conducted in Sweden. We have considered different options, but have found that adding an explicit mention of the exploratory approach does not make the title more informative, but rather makes it more difficult to read (e.g., “Exploring the relationships between self-reported impact of the COVID-19 pandemic, affective responding, and subjective well-being: A Swedish survey”). Therefore, we have chosen to retain the original title. However, to make the exploratory approach more salient we have changed the word “examined” to “explored” in the following sentence in the Abstract: “The current research explored the relationships between self-reported affective responding, perceived personal consequences of the COVID-19 pandemic, and subjective well-being.”

2. In the introduction, the authors could mention other works on the validation of measurement instruments that have recently been developed to evaluate different aspects related to the Covid-19 pendemic. For example, Questionnaire on Perception of Threat from COVID-19.

Because of space concerns, and to retain focus on the central aspects of our study, we have not been able to incorporate a detailed discussion on the validation of other instruments. Instead, we have added a summary statement in the second paragraph of the Introduction to this effect, and cite the suggested Questionnaire on Perception of Threat from COVID-19 as an example: “Moreover, because of its extraordinary global impact, rapid attempts have been made to develop instruments specifically tailored to capturing the unique psychological consequences and perceptions of COVID-19 (e.g., Ahorsu et al., 2020; Pérez-Fuentes et al., 2020).”

3. It is also recommended to check that new references have not been included in the discussion that, previously, were not cited in the introduction section.

Especially because of the exploratory nature of the present work, interpretation of the data is facilitated by incorporating previously-unaddressed literature into the discussion. We want to avoid giving readers the impression that we anticipated the results by preemptively introducing ideas raised by the results themselves. This approach is consistent with recent discussions and recommendations concerning the framing and presentation of research (see, e.g., Giner-Sorolla, 2012).

Reviewer #2

1. The introduction is quite loosely related to the main purpose of the study. Although the general aim of the study – searching for risk factors of worsened adaptation to the pandemic – is justified, the sentences indicating a lack of hypotheses are confusing and methodologically doubtful. If Author(s) did not have any hypotheses, how they determine the independent variables? The design of the study indicates that the Authors were investigating anger, irritability and impulsive behaviors, moderator role of emotion regulation strategies and the relatively new construct: self-reported impact of the COVID-19 pandemic. Thus, in the introduction these variables should be not only included but also put in the context of the psychological responding to the pandemic. Why studying anger and related constructs should be useful in the context o the pandemic? Which is the role of restriction due to the pandemic? How about the particular Swedish context in this matter? In my opinion the introduction should be rewritten in order to include the hypotheses or a model of the study.

We agree that the Introduction is not closely tied to the specific model tested in our data analyses. The reason for this is that we did not have a specific model in mind before collecting the data. Instead, the specification of the model developed as we explored the data. Therefore, we do not think it would be an accurate depiction of the research process if we rewrote the Introduction to suggest that the study was designed to test a particular model. It is true, however, that we had preliminary ideas about variables that may plausibly be considered predictors of the perceived impact of the COVID-19 pandemic. Furthermore, we agree with the reviewer that the reasons for exploring these particular variables as predictors were not adequately articulated in the original submission. We have therefore added a new paragraph on page 6, where we introduce trait anger, impulsivity, and irritability and explain why they may be relevant in the current context. In that same paragraph, we also clarify that these variables were part of a battery of instruments used for a different study on the measurement of irritablility. Thus, data on these variables were not collected primarily for the purpose of the current study, but may nonetheless be relevant for understanding individual differences in the perceived personal impact of the COVID-19 pandemic.

Moreover, we have added a brief rationale as to why Sweden is an interesting case given its rather unique governmental response (p. 5): “The comparatively liberal governmental response in Sweden makes the current sample an important complement to previous studies conducted in areas with more restrictive measures in place.”

Finally, we have rewritten the sentence which stated that we did not have any hypotheses. While the new sentence maintains that predictions had not been made prior to data collection, we emphasize that we were particularly interested in the moderating role of emotion regulation strategies (p. 6): “Although we had not made specific predictions prior to the collection of the data, we were particularly interested in exploring whether emotion regulation strategies play a protective or aggravating role regarding the relationship between self-reported impacts of the COVID-19 pandemic and subjective well-being.”

2. I think that the Authors can include an exemplary attention check in order to make clear why relatively large portion of the participants was dropped out from the analysis.

We have included an example of the options presented to participants in the attention checks (p. 11). 

3. Was the self-reported impact of the COVID-19 hypothesized as a unifactorial variable? At the beginning of the results section the Authors described results separately for each item of the self-reported impact, but later on factor analyses them and conducted analyses on the two factors. In my opinion, the factor analysis should be presented from the very beginning and the rest analyses could be conducted regarding the two factors of the self-reported pandemic impact.

The present project was exploratory, so we did not have any particular hypotheses concerning the factor structure of the self-reported COVID-19 impact items. The factor analysis was conducted specifically to reduce the dimensionality of the six items. That is, it was a largely pragmatic decision, with the goal of reducing the number of variables in the subsequent moderation analyses. We are now more explicit about this in the text.

Prior to the factor analysis, we believe the use of the individual items is more informative, since it provides more nuance. For example, impacts specifically on family life may be related to the reported increases in domestic violence during the pandemic. These more fine-grained interpretations are more difficult to discuss when examining only factor scores, though the composites are convenient for analyses that involve more complex models, like the moderation analyses.

4. In the discussion try to avoid using technical terms like: factor 1 and factor 2 – please use a labels of these factors.

We have now removed references to “Factor 1” and “Factor 2” in the discussion section. Instead, we refer to their constituent items.

5. Since the demographical variables are not the central independent variables in the study, I think that their role in prediction of the COVID-19 impact should be described more concisely (e.g. without separate sections, but only in descriptive statistics section).

We have moved the report on correlations with demographic variables to a supplemental document on OSF. The only exception is the relationship between impacts and occupation, which is the only significant correlation between a demographic variable and self-reported COVID-19 impacts. The section on demographics has now been reduced to a single paragraph, which we believe more concisely articulates the important results while also allowing interested readers to easily find more detailed information online.

6. While investigating the moderation, I suggest the Author(s) to focus on the particular method (SEM or regression analysis). 

We have now revised the moderation section to place focus on the OLS regression analysis, rather than the latent variable approach, which has been reframed as a robustness check on the more conventional regression. Although we think the latent variable approach has many advantages, the OLS regression is more parsimonious and arrives at the same basic conclusions. We think readers will find this approach more easily digestible.

7. The figure attached to the manuscript seems to be distorted (when compared to the figure in the osf project). 

We have corrected the figure both in the manuscript and in the online repository.

8. The power analysis in the manuscript was made for correlations analyses. However, the Author(s) may also included post-hoc power analysis for the interaction detection, e.g. using delta f2 (change of the explained variance between steps in hierarchical regression; please remember to included statistics for regression model when describing the interaction: F, adj. R2, change in R2 due to interactions included in the model, etc.).

Consistent with arguments by statisticians and data analysists (see, e.g., Hoenig & Heisey, 2001), we believe post-hoc power analyses to be generally uninformative, especially when conducted in relation to the effects observed in a study. However, we already report a sensitivity analysis that provides the standardized effect sizes for which the sample size gives 80% and 99% power. These effects are presented as correlation coefficients, which can easily be converted into other standardized metrics, such as f-squared.

References

Giner-Sorolla, R. (2012). Science or Art? How Aesthetic Standards Grease the Way Through the Publication Bottleneck but Undermine Science. Perspectives on Psychological Science, 7(6), 562–571. https://doi.org/10.1177/1745691612457576

Hoenig, J. M., & Heisey, D. M. (2001). The Abuse of Power: The Pervasive Fallacy of Power Calculations for Data Analysis. The American Statistician, 55(1), 19–24. https://doi.org/10.1198/000313001300339897

---

## [Decision Letter · Decision Letter 1]

20 Jul 2021

PONE-D-21-05122R1

Self-reported impact of the COVID-19 pandemic, affective responding, and subjective well-being: A Swedish survey

PLOS ONE

Dear Dr. Gröndal,

Thank you for submitting your manuscript to PLOS ONE. After careful consideration, we feel that it has merit but does not fully meet PLOS ONE’s publication criteria as it currently stands. Therefore, we invite you to submit a revised version of the manuscript that addresses the points raised during the review process.

We look forward to receiving your revised manuscript.

Kind regards,

Valerio Capraro

Academic Editor

PLOS ONE

Journal Requirements:

Additional Editor Comments (if provided):

One of the reviewers has some minor comments before publication. Please address these last comments at your earliest convenience. I am looking forward for the final version.

Reviewers' comments:

Reviewer's Responses to Questions

**Comments to the Author**

1. If the authors have adequately addressed your comments raised in a previous round of review and you feel that this manuscript is now acceptable for publication, you may indicate that here to bypass the “Comments to the Author” section, enter your conflict of interest statement in the “Confidential to Editor” section, and submit your "Accept" recommendation.

Reviewer #2: All comments have been addressed

2. Is the manuscript technically sound, and do the data support the conclusions?

Reviewer #2: Yes

3. Has the statistical analysis been performed appropriately and rigorously? 

Reviewer #2: Yes

4. Have the authors made all data underlying the findings in their manuscript fully available?

Reviewer #2: Yes

5. Is the manuscript presented in an intelligible fashion and written in standard English?

Reviewer #2: Yes

6. Review Comments to the Author

Reviewer #2: After reading the extensive review of the previous version of the manuscript and the thorough answers of the Authors I have only some minor suggestions or questions:

p. 17; l. 351-259: The Authors may introduce names of Factor 1 and 2 of the self-reported COVID-19 impact. It seems that Factor 1 refers to self-reported COVID-19 impact on free time activities, while the factor 2 refers to self-reported COVID-19 impact on work and family life. The analysis left also unanswered questions about the moderating role of ERQ strategies for associations between irritability and self-reported COVID-19 impact. It could be mentioned in the future analyses that individuals with some personality factors that may have been predictive for experiencing grater burden of the pandemic (e.g. irritability) may be protected by efficient emotion regulation.

p. 18; Table 2: Did the Authors check whether the entered interaction term resulted in significant change in R squared of the regression model?

Figure 1: Please, provide more precise names of the axes: ERQ R could be written as Reappraisal, while COVID-19 Factor 1 (ustandardized b) – if I understand it correctly it should be: Association between self-reported Covid-19 impact (Factor 1) on SWLS (unstandardized b).

7. PLOS authors have the option to publish the peer review history of their article (what does this mean?). If published, this will include your full peer review and any attached files.

Reviewer #2: No

---

## [Author Response · Author response to Decision Letter 1]

3 Sep 2021

Dear professor Capraro,

Editor comments

We have reviewed the reference list to make sure that it is complete and correct. Moreover, we have found that none of the entries in the list have been retracted.

Reviewer #2 comments

1. p. 17; l. 351-259: The Authors may introduce names of Factor 1 and 2 of the self-reported COVID-19 impact. It seems that Factor 1 refers to self-reported COVID-19 impact on free time activities, while the factor 2 refers to self-reported COVID-19 impact on work and family life. The analysis left also unanswered questions about the moderating role of ERQ strategies for associations between irritability and self-reported COVID-19 impact. It could be mentioned in the future analyses that individuals with some personality factors that may have been predictive for experiencing grater burden of the pandemic (e.g. irritability) may be protected by efficient emotion regulation.

It is only one section of the results where the labels Factor 1 and Factor 2 are used as predictors in models. As such, we don’t think it’s worth introducing a name for these indexes, both because of the brevity of that section and also to avoid the naming fallacy. However, we agree that some additional clarity about these factors may be useful, so we have added some parentheticals to the section to make it especially clear what items the factors comprise. Moreover, in the Discussion, we refer to the factors in terms of their actual content (e.g., “The reported impact of the pandemic on family life, work/study, and finances…”) rather than factor numbering.

We are hesitant to add any further analyses to the paper, as it already contains a considerable amount of results. However, in response to this comment, we examined the possibility that irritability interacts with the self-reported COVID-19 impacts. Regression analyses suggested no such interaction. 

2. p. 18; Table 2: Did the Authors check whether the entered interaction term resulted in significant change in R squared of the regression model?

We have revised out presentation of the regression analysis to include a model selection procedure that assesses change in model fit from adding the interaction terms. We now present more parsimonious models based on this procedure, which support substantively the same conclusions as the previously presented models. It is clear from this presentation that the introduction of the critical interaction term significantly improved model fit.

3. Figure 1: Please, provide more precise names of the axes: ERQ R could be written as Reappraisal, while COVID-19 Factor 1 (ustandardized b) – if I understand it correctly it should be: Association between self-reported Covid-19 impact (Factor 1) on SWLS (unstandardized b).

We have edited the labels of Figure 1 as suggested.

---

## [Editor Report · Decision Letter 2]

6 Oct 2021

Self-reported impact of the COVID-19 pandemic, affective responding, and subjective well-being: A Swedish survey

PONE-D-21-05122R2

Dear Dr. Gröndal,

We’re pleased to inform you that your manuscript has been judged scientifically suitable for publication and will be formally accepted for publication once it meets all outstanding technical requirements.

Kind regards,

Valerio Capraro

Academic Editor

PLOS ONE
---

## [Editor Report · Acceptance letter]

8 Oct 2021

PONE-D-21-05122R2 

Self-reported impact of the COVID-19 pandemic, affective responding, and subjective well-being: A Swedish survey 

Dear Dr. Gröndal:

I'm pleased to inform you that your manuscript has been deemed suitable for publication in PLOS ONE. Congratulations! Your manuscript is now with our production department. 

Kind regards, 

on behalf of

Dr. Valerio Capraro 

Academic Editor

PLOS ONE